# Structure and optical properties of perovskite-embedded dual-phase microcrystals synthesized by sonochemistry

Sangyeon Cho [1,2] & Seok Hyun Yun [1,2]*

Cesium lead halide perovskite ($CsPbX_3$, X = Cl, Br, I) nanocrystals embedded in $Cs_4PbX_6$ or $CsPb_2X_5$ matrices have received interests due to their excellent optical properties. However, their precise endotaxial structures are not known, and the origin of photoluminescence remains controversial. Here we report a sonochemistry technique that allowed us to synthesize high-quality $CsPbBr_3$-based microcrystals in all ternary phases, simply by adjusting precursor concentrations in a polar aprotic solvent, N,N-dimethylformamide. The microcrystals with diverse morphologies enabled us to visualize the lattice alignments in the dual-phase composites and confirm $CsPbBr_3$ nanocrystals being the photoluminescent sites. We demonstrate high solid-state quantum yield of >40% in $Cs_4PbBr_6$/$CsPbBr_3$ and lasing of $CsPbBr_3$ microcrystals as small as 2 μm in size. Real-time optical analysis of the reaction solutions provides insights into the formation and phase transformation of different $CsPbBr_3$-based microcrystals.

[1] Wellman Center for Photomedicine, Massachusetts General Hospital and Harvard Medical School, Cambridge, MA 02139, USA. [2] Harvard-MIT Health Sciences and Technology, Massachusetts Institute of Technology, Cambridge, MA 02139, USA. *email: syun@hms.harvard.edu

Three-dimensional (3D) lead halide perovskites (LHPs) with the form of $APbX_3$ (A=$Cs^+$, $CH_3NH_3^+$, X=$Cl^-$, $Br^-$, $I^-$) are promising optical materials. These materials offer a long carrier lifetime (>1 µs), long exciton diffusion length[1,2] (>1 µm), large optical cross-sections[3] (~$10^{-13}$ $cm^2$), and defect tolerance owing to the antibonding character of the conduction and valence bands[4]. These properties make them an attractive building block for solar cells, light-emitting diodes, and lasers[5]. Among various types of LHPs, all-inorganic $CsPbBr_3$ received increasing interest due to their high luminescent quantum yields in solid states in the green range and superior environmental stability to other perovskites with organic cations. $CsPbBr_3$ has two lower-dimensional counterparts: zero-dimensional (0D) $Cs_4PbBr_6$ and quasi-two-dimensional (2D) $CsPb_2Br_5$. They represent different ternary phases of the Cs–Pb–Br compounds and can be formed from precursors, such as CsBr and $PbBr_2$ (Supplementary Fig. 1)[6]. $Cs_4PbBr_6$ and $CsPb_2Br_5$ are known to have large bandgap energies of 3.7 eV and 3.1 eV, respectively[7,8]. Nonetheless, some confusion has arisen when these non-perovskite materials were claimed to generate green photoemission[4,9]. Recent studies[7,10–12] have suggested that these materials contain $CsPbBr_3$ nanocrystals (NCs) that are responsible for photoluminescence.

We refer these dual-phase materials to as $Cs_4PbBr_6$/$CsPbBr_3$ ($CsPbBr_3$ NCs in a $Cs_4PbBr_6$ matrix) and $CsPb_2Br_5$/$CsPbBr_3$ ($CsPbBr_3$ NCs in a $CsPb_2Br_5$ matrix). While much progress in the dual-phase materials is expected, it has been difficult to visualize the relative lattice orientation between LHPs NCs and the host matrix clearly by using HRTEM, due to difficulties such as the low damage threshold of the materials by electron beam and the inadequate sample sizes being too small lateral size (<10 nm) to observe more than four different lattice planes at given electron beam or too large in thickness (larger than few hundreds of nm) in thickness to get clear images[7,12–14].

Herein, we report a new method based on sonochemistry that enables a facile, rapid synthesis of various phase, and dimensional $CsPbBr_3$ perovskites microcrystals in a polar aprotic solvent, N,N-dimethylformamide. We show this technique can produce both types of dual-phase materials, as well as single-phase LHP microcrystals, with various surface morphologies depending on precursor concentrations. Our investigation provides insights into the formation kinetics and phase transition of the Cs–Pb–Br compounds. The produced microparticles enabled us to investigate the lattice structures and optical properties of the various $CsPbBr_3$-based compounds.

## Results and discussion

**Sonochemical synthesis of various LHPs.** The general scheme of sonochemical synthesis starts with placing two precursor salts, CsBr and $PbBr_2$, with sufficient quantities beyond their maximum soluble amount in a polar aprotic solvent. For CsBr and $PbBr_2$ salts, N,N-dimethylformamide (DMF) produced high-quality $CsPbBr_3$ microparticles among different polar aprotic solvents having similar dipole moments (acetone, ethyl acetate (EtoAC), γ-butyrolactone (GBL), and dimethyl sulfoxide (DMSO)) (Supplementary Fig. 2). So, all the experiments presented herein after were obtained using DMF. As illustrated in Fig. 1a, after several minutes of ultrasonication at room temperature (Supplementary Fig. 3), the salts are fully dissolved and produce a stable solution. The final solution varies in color, depending on the ratio of the starting concentration of the precursors (Supplementary Fig. 4).

Figure 1b summarizes our finding arranged in a two-dimensional phase diagram, along with the SEM images of various distinct types of microparticles formed. We used parameters "a" and "b" to denote the concentrations of the CsBr and $PbBr_2$ precursors, respectively. The values are normalized to 75 mM (e.g., "a = 1" corresponds to 75 mM of CsBr, and "b = 2" refers to 150 mM of $PbBr_2$).

From stoichiometry, the ideal concentration ratio to produce $CsPbBr_3$ would be a/b = 1, if the two precursor materials were equally dissolved in the solvent. Experimentally, $CsPbBr_3$ microcrystals were produced when the ratio a/b was approximately between 0.8 and 2, and both "a" and "b" are in a range from 1 to 4 (i.e., 75–300 mM). The orange-color solution obtained after ultrasonication contains single-phase $CsPbBr_3$ microparticles with the cuboidal shape (Supplementary Fig. 5).

When a/b <0.8 and b >1, the product of reaction is dual-phase $CsPb_2Br_5$/$CsPbBr_3$ composites, which show intense yellow color under room light. Their surface morphology varied depending on the precursor concentration (Supplementary Fig. 6). For a = 1 (75 mM) and b = 2 or 3 (75 or 150 mM), the microparticles have largely octahedral shapes. With b = 4 (300 mM), the particles tend to have irregular shapes and rough surfaces, probably resulting from rapid surface nucleation due to the high $PbBr_2$ concentration. No crystals were formed at low concentrations of a = 0.5 (37.5 mM) and b = 1 (75 mM).

When a/b >2 and a >2 (150 mM), we found that single-phase $CsPbBr_3$ microcrystals are initially formed, but converted to white-color $Cs_4PbBr_6$ microcrystals and then to lemon-color dual-phase $Cs_4PbBr_6$/$CsPbBr_3$ composites. $Cs_4PbBr_6$/$CsPbBr_3$ composites are found in a mixture of rhombus and hexagonal plates (Supplementary Fig. 7). When these $Cs_4PbBr_6$/$CsPbBr_3$ plates are left for about 1 h in the solution with ultrasonication being off, they undergo phase transition to $Cs_4PbBr_6$/$CsPbBr_3$ with micro-discoidal shapes. The same morphological transition was observed when the solution was vigorously shaken by hands. This dynamic process is described later in more detail.

When a/b = 2 and a <= 2 (150 mM), $CsPbBr_3$ microparticles are initially formed and converted to $Cs_4PbBr_6$ microcrystals, and they remain as the final product. Single-phase $Cs_4PbBr_6$ microcrystals have a granular structure (Supplementary Fig. 8).

**Structures of dual-phase LHPs.** To identify the crystal structure and stoichiometry of the various products, we performed powder X-ray diffraction (PXRD) and energy dispersive X-ray spectroscopy (EDS). The data (Supplementary Figs. 9–12) confirmed the orthorhombic structure of $CsPbBr_3$ (space group *Pbnm*, A = 8.20 Å, B = 8.24 Å, C = 11.74 Å), the trigonal structure of $Cs_4PbBr_6$ in the single- and dual-phase $Cs_4PbBr_6$ products ($R\bar{3}c$, A = B = 13.73 Å, C = 17.32 Å), and the tetragonal structure of $CsPb_2Br_5$ in the dual-phase composite (*I4/mcm*, A = B = 8.45 Å, C = 15.07 Å). These results are consistent with previous reports[8,15,16].

Unlike previous dual-phase materials[7,12–14], structurally anisotropic microparticles produced by sonochemistry were well suited to obtain high-quality HRTEM having more than four different lattice planes at given the direction of an electron beam. For HRTEM imaging, we used micro-discoidal $Cs_4PbBr_6$/$CsPbBr_3$ (a = 3 (225 mM), b = 1 (75 mM)) having average thickness of 80 nm (N = 15) and wedding-cake $CsPb_2Br_5$/$CsPbBr_3$ (a = 1 (75 mM), b = 4 (300 mM)) consisting of multiple thin layers having average thickness of 50 nm (N = 15) (Supplementary Fig. 13).

The real-space images and the corresponding fast Fourier transform (FFT) analysis revealed the relative lattice orientation of the endotaxial structures (Fig. 2a, c; Supplementary Fig. 14). In the case of $Cs_4PbBr_6$/$CsPbBr_3$, well-defined lattice fringes with 3.7 Å and 4.2 Å having intersection angle of 42° are indexed to (212) and (210) of the $Cs_4PbBr_6$ matrix, and higher-contrast lattice fringes with 3.7 Å and 3.5 Å having intersection angle of 67° are indexed to (210) and (021) of the $CsPbBr_3$ NCs. This suggests that the [210]

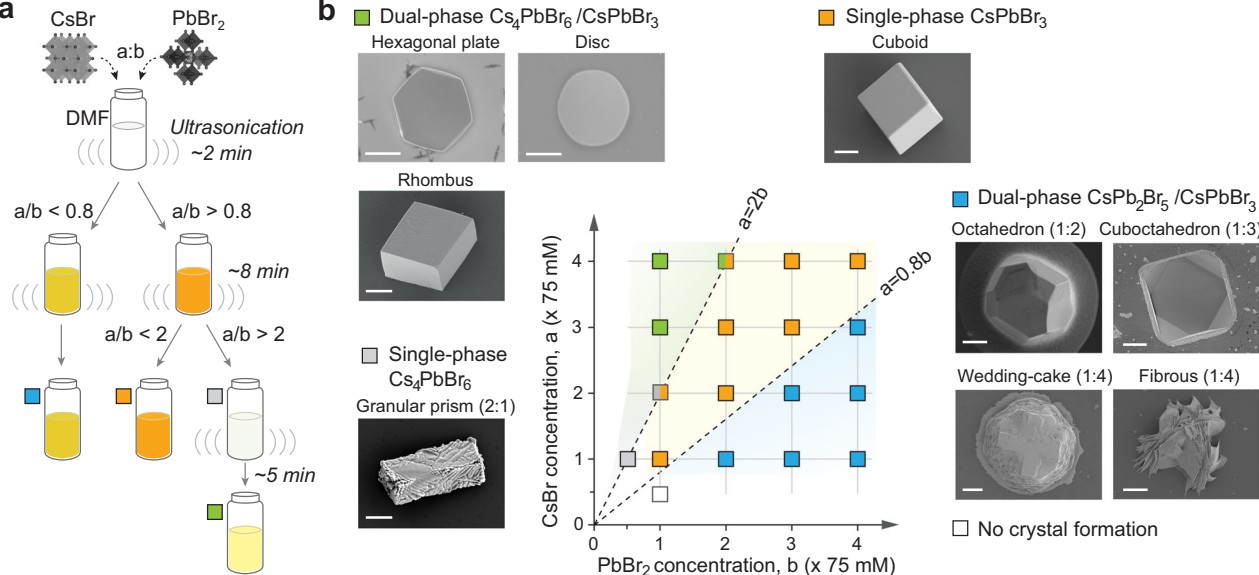

**Fig. 1 Sonochemical synthesis of various CsPbBr₃-based microcrystals. a** Schematic of the sonochemical synthesis using ultrasonication of CsBr and PbBr₂ salts in DMF. The starting concentration of the precursor materials are denoted as "*a*" and "*b*". **b** Two-dimensional concentration phase diagram of the sonochemical reaction products. SEM images of dual-phase Cs₄PbBr₆/CsPbBr₃ (hexagonal plate, microdisc, and rhombus), single-phase Cs₄PbBr₆ (granular prism), single-phase CsPbBr₃ (cuboid), dual-phase CsPb₂Br₅/CsPbBr₃ (truncated octahedron, cuboctahedron, wedding cake, and fibrous). Scale bars, 2 μm.

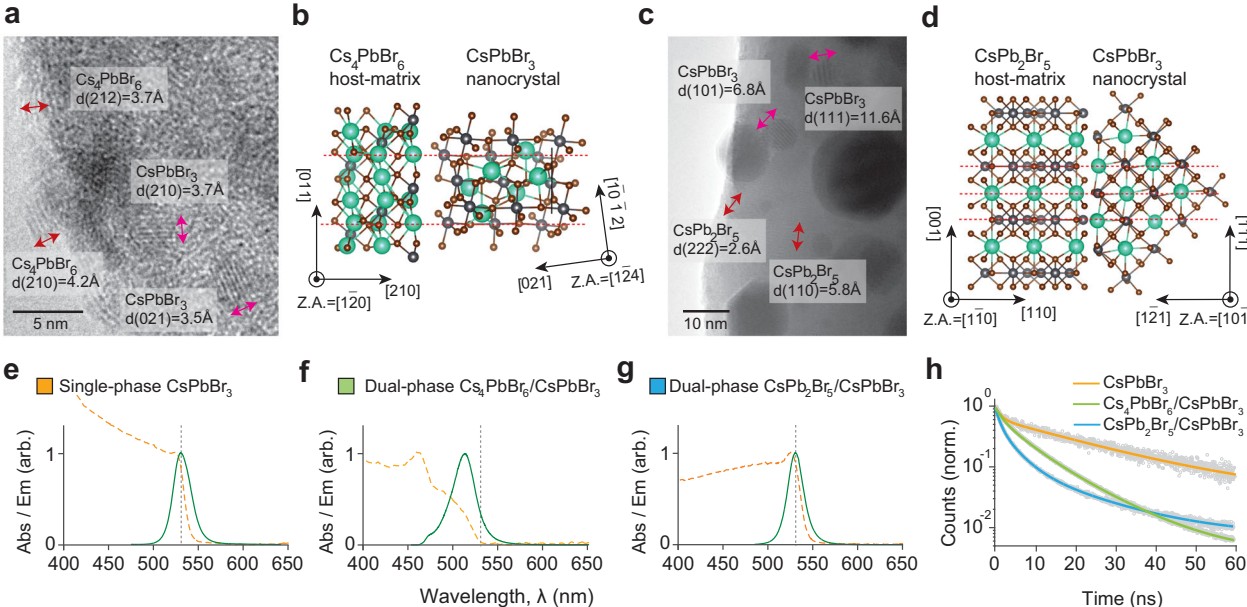

**Fig. 2 Structural and optical characterization of the CsPbBr₃-based materials. a** HRTEM image of a dual-phase Cs₄PbBr₆/CsPbBr₃ microdisc. **b** Cs₄PbBr₆ and CsPbBr₃ crystal structure based on the HRTEM image. **c** HRTEM image of a dual-phase CsPb₂Br₅/CsPbBr₃ wedding-cake crystal. **d** CsPb₂Br₅ and CsPbBr₃ crystal structure based on the HRTEM image. **e**–**g** Absorbance (dashed orange lines) and fluorescence (solid green lines; excitation at 480 nm) spectra of CsPbBr₃, Cs₄PbBr₆/CsPbBr₃, and CsPb₂Br₅/CsPbBr₃. **h** Time-resolved photoluminescence measurement. The measured data (circles) are fitted with triple exponential curves (lines).

axis of Cs₄PbBr₆ is tilted by 6° with respect to the [021] axis of CsPbBr₃. For CsPb₂Br₅/CsPbBr₃, low-contrast lattice fringes with 2.6 Å and 5.8 Å having intersection angle of 35° are indexed to (222) and (110) of the CsPb₂Br₅ matrix, and lattice fringes in darker sub-regions with 6.8 Å and 11.6 Å having intersection angle of 35° are indexed to (210) and (021) of the CsPbBr₃ NCs. Hence, the [110] axis of CsPb₂Br₅ is aligned to the [1$\bar{2}$1] axis of CsPbBr₃. A computational model based on the data confirmed good facet

matching between CsPbBr₃ NCs and non-perovskite matrices (Fig. 2b, d). From the TEM images, we determined the effective size of CsPbBr₃ NCs embedded in the matrices by measuring the diameter of the largest circle circumscribing the NCs (Supplementary Fig. 15). The CsPbBr₃ NCs in Cs₄PbBr₆ have sizes of 3–5 nm with a mean effective diameter of 4.2 nm. The CsPbBr₃ NCs in CsPb₂Br₅ are larger, ranging from 10 to 20 nm, with a mean effective diameter of 14.6 nm.

**Optical properties of dual-phase LHPs.** Using a custom-built microscope coupled with a grating-based spectrometer (Supplementary Fig. 16), we measured the optical emission and absorption spectra of various product particles either in solution (Fig. 2e–g). The optical spectra did not change after the particles have been transferred to a glass substrate. $CsPbBr_3$ microcrystals have an absorption edge at 538 nm (2.31 eV), weak excitonic peak at 523 nm, and low Urbach energy of 23 meV. Dual-phase $Cs_4PbBr_6/CsPbBr_3$ microcrystals have an absorption edge at 525 nm (2.37 eV), and their fluorescence peaks are blue-shifted to 512 nm. The magnitude of blue shift varied between 10 and 21 nm, depending on the precursor ratio and the conversion method (Supplementary Fig. 17). The bandgap changes, $\Delta E$, by quantum confinement is given by:

$$\Delta E \approx \frac{\hbar^2\pi^2}{2d^2}\left(\frac{1}{m_e^*} + \frac{1}{m_h^*}\right) - \frac{1.8\,e^2}{4\pi\varepsilon d} \quad (1)$$

where $m_h^* = 0.14$ and $m_e^* = 0.15$ denote the effective mass of the hole and electron[17], respectively, in $CsPbBr_3$ in the unit of the electronic mass, $d$ the diameter of a spherical potential well, $\varepsilon$ the permittivity of the matrix surrounding $CsPbBr_3$ NCs. $CsPbBr_3$ NCs. $\varepsilon/\varepsilon_0 = 3.1$ was calculated for $Cs_4PbBr_6$ by a density functional theory[18]. The spectral shifts we measured from the spectra indicate $d = 6.2$ nm and 5.6 nm, respectively. These values are reasonable, but larger than the mean diameter of 4.2 nm. The discrepancy may be attributed to the nonspherical shapes of the nanocrystals and interfacial effects with the $Cs_4PbBr_6$ matrix. The dielectric constant of $CsPb_2Br_5$ matrix is unknown. Assuming it is the same as $Cs_4PbBr_6$, the quantum confinement effect for NCs with the mean diameter of 14.6 nm is estimated to be $-6$ meV. The fluorescence peak of $CsPb_2Br_5/CsPbBr_3$ microparticles is at 530 nm, ~1 nm shifted from the 531 nm peak of $CsPbBr_3$ microcrystals. This shift of ~4.4 meV corresponds to $d = 11$ nm.

We investigated time-resolved photoluminescence using a picosecond frequency-doubled laser ($\lambda = 382$ nm). The experimental time-resolved photoluminescence data (Fig. 2h; Supplementary Table 1) were fitted to a three-exponential decay curve:

$$f(t) = A_1 e^{-\frac{t}{\tau_1}} + A_2 e^{-\frac{t}{\tau_2}} + A_3 e^{-\frac{t}{\tau_3}} \quad (2)$$

where $A_1$, $A_2$, and $A_3$ are pre-exponential factors, and $\tau_1$, $\tau_2$, and $\tau_3$ are lifetime constants. The total decay time was computed from weighted lifetime constants:

$$\tau_{tot} = A_1\tau_1 + A_2\tau_2 + A_3\tau_3 \quad (3)$$

The radiative decay time of the sample is related to total decay time and absolute PLQY:

$$\tau_{rad} = \tau_{tot} * PLQY \quad (4)$$

Dual-phase $Cs_4PbBr_6/CsPbBr_3$ has a much faster radiative lifetime of 9.7 ns, compared with the lifetime of 1.2 µs for single-phase $CsPbBr_3$ and 4.4 µs for dual-phase $CsPb_2Br_5/CsPbBr_3$. $Cs_4PbBr_6/CsPbBr_3$ has a high photoluminescence quantum yield (PLQY) of over 40% due to both the quantum confinement and low dielectric constant of $Cs_4PbBr_6$[13].

**Lasing of optically pumped LHP microcrystals.** Single-mode lasing from single-phase $CsPbBr_3$ microcrystals was observed when excited by nanosecond-pulsed optical pumping at 480 nm (Fig. 3; Supplementary Fig. 18). The smallest size of lasing $CsPbBr_3$ microcrystals was 2 µm (Fig. 3b; Supplementary Fig. 18). The laser emission linewidth was ~0.2 nm above a threshold pump energy of 1.7 mJ/cm², and the spontaneous emission factor (β) was 0.05 (Fig. 3c). The other device presents the laser emission linewidth of 0.3 nm above a threshold pump energy of

2.2 mJ/cm², and the spontaneous emission factor was $10^{-3}$ (Supplementary Fig. 18). The converted threshold pump energy to threshold carrier density is $\sim 3 \times 10^{19}$ cm$^{-3}$, which is higher than theoretical estimation of Mott density ($10^{18}$ cm$^{-3}$)[19]. This infers lasing in the electron hole plasma state (EHP), rather than excitonic state[5,19], which is beneficial to build up large population inversion via bandgap renormalization (BGR)[20]. The laser emission of single-phase $CsPbBr_3$ microparticles in air (15 samples) at a pump fluence twice the lasing threshold was prolonged for $10^5$ pump pulses (5000 s at 20 Hz) with a pulse-to-pulse wavelength fluctuation of 0.47 nm (Supplementary Fig. 18). On the contrary, dual-phase microparticles did not support laser oscillation even at higher pump energy levels up to tens of mJ/cm². Considering the high PLQY of $CsPbBr_3$ NCs in the dual-phase composites, we attribute the failure to reach lasing threshold to the weaker cavity resonance due to the lower refractive index (~1.8) of the $Cs_4PbBr_6$[21] and $CsPb_2Br_5$ matrices compared with the index (~2.6) of $CsPbBr_3$[5], and relatively small amount of the optical gain in the excitonic state due to un-normalized bandgap.

**Mechanism of dual-phase formation.** To gain insights into the mechanism of dual-phase formation, we investigated distinct intermediate reaction steps, which involves color changes of the solution. The sonochemical synthesis of $Cs_4PbBr_6/CsPbBr_3$ composite is comprised six distinct steps: the formation of orange-color $CsPbBr_3$ (steps i–iv), the phase transformation from orange-color $CsPbBr_3$ to white-color single-phase $Cs_4PbBr_6$ (step v), and the formation of dual-phase $Cs_4PbBr_6/CsPbBr_3$ (step vi) (Fig. 4a; Supplementary Movie 1).

Our interpretation of the process is as follows. When CsBr and $PbBr_2$ salts were mixed with DMF with concentration of $a = 3$ and $b = 1$, orange-color $CsPbBr_3$ layer is immediately formed via interfacial conversion on the surface of undissolved salts [step i]. The ultrasonic pressure and temperature modulation allows the remaining salts to be completely dissolved [step ii]. Simultaneously, the reaction intermediate, $PbBr_4^{2-}$, increases via:

$$PbBr_2(s) + 2Br^-(sol) \rightarrow PbBr_4^{2-}(sol)\,[step\ iii]$$

Since the reaction species are optically transparent, the solution turns clear. The spontaneous nucleation and growth of $CsPbBr_3$ occur when the concentration of $PbBr_4^{2-}$ reaches the level of saturation, at which the solution turns orange. The crystallization reaction may be described as:

$$Cs^+ + PbBr_4^{2-} \rightarrow CsPbBr_3(s) + Br^-(sol)\,[step\ iv]$$

We measured the time trace of the color intensity (steps i–iv) (Supplementary Fig. 19), and calculated the reaction rates from the slope and duration of the color intensity profile. Figure 4b shows logarithmic plots of the reaction rates, $v$, as a function of the reciprocal of temperature $T$, and an overall reaction coordinate diagram. From the curve fitting of the data with the Arrhenius and Eyring equation, we obtained $E_a = -35$ kJ/mol for step (ii), 27 kJ/mol for step (iii), and 23 kJ/mol for step (iv). The negative activation energy and exothermic dissolution of bulk $CsPbBr_3$ in step (ii) are due to low lattice-formation energy of $CsPbBr_3$. This low energy barrier is a double-edged sword making LHPs easy to be crystallized and degraded[22]. The measured activation energies of the step (iii) and step (iv) are approximately three times smaller than that of conventional thin film formation (86 kJ/mol)[23]. The reduced activation energy likely comes from the vibrant oscillation of the pressure and temperature in ultrasonication microbubbles[24]. This low activation energy is a key to the rapid synthesis.

To understand the transformation from intermediate single-phase $CsPbBr_3$ to single-phase $Cs_4PbBr_6$, we stopped ultrasonication after 10 min right after step (iv), transferred a titer amount of

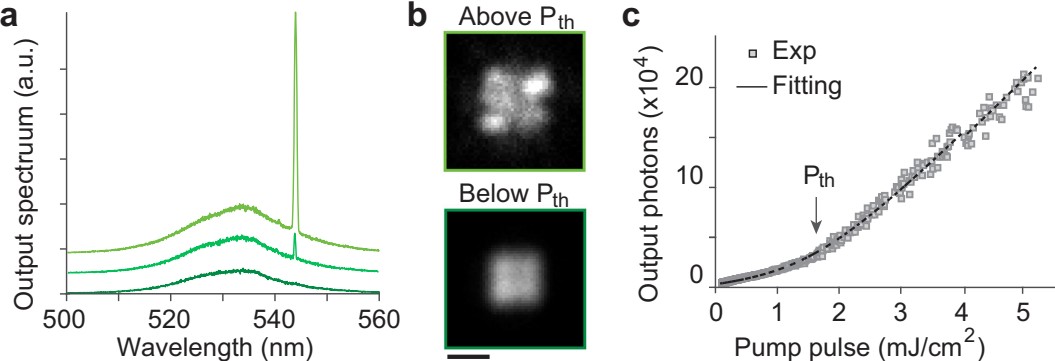

**Fig. 3 Lasing from single-phase CsPbBr₃ microcrystals. a** Output spectra from a 2-μm sized $CsPbBr_3$ microcrystal upon nanosecond optical pumping (480 nm) below and above lasing threshold. **b** Wide-field fluorescence images below and above laser threshold. Scale bar, 2 μm. **c** A light-in–light-out curve, showing a threshold pump fluence of 1.7 mJ/cm² and a spontaneous emission factor (β) of ~0.05.

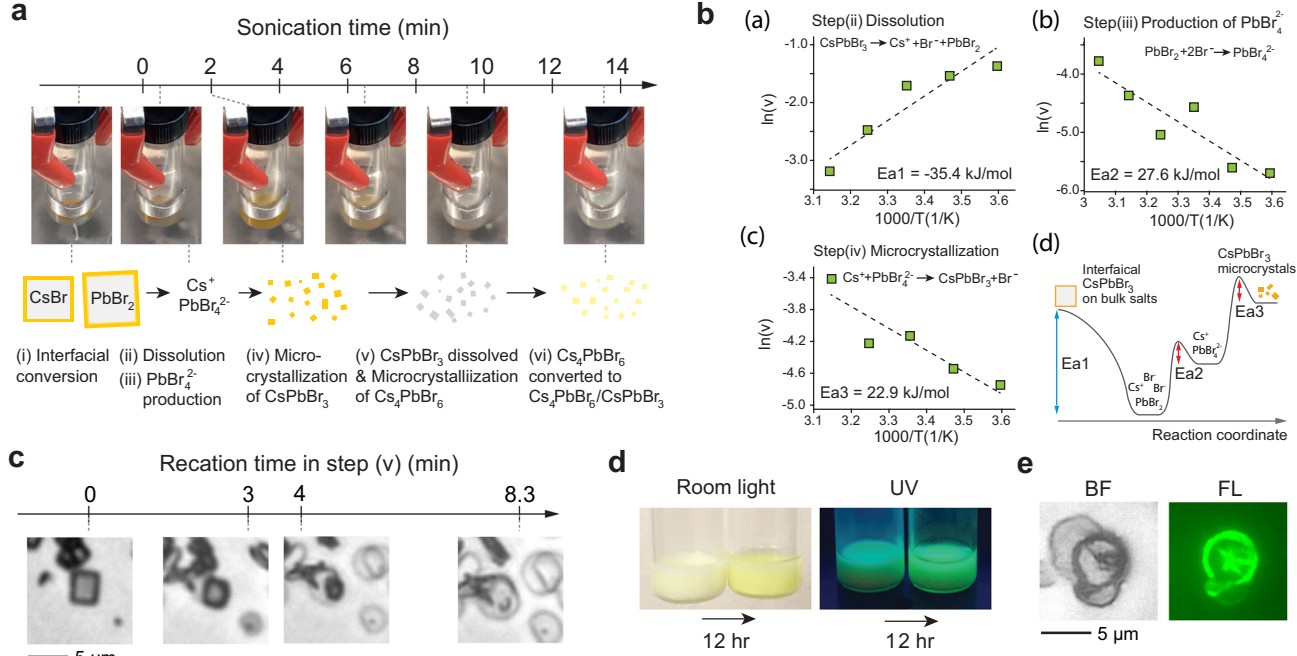

**Fig. 4 Phase transformation from single-phase CsPbBr₃ to dual-phase Cs₄PbBr₆/CsPbBr₃. a** Time-lapse change of precursor solution ($a = 3$, $b = 1$) during ultrasonication and a schematic of six reaction intermediate steps. **b** (a–c) The temperature dependence of the reaction rate, $v$ [s⁻¹], for each reaction step (ii to iv); and (d) a reaction coordinate diagram of the sonochemical synthesis of $CsPbBr_3$ microcrystals. **c** Bright-field images showing the phase transformation of single-phase $CsPbBr_3$ cuboids to single-phase $Cs_4PbBr_6$ microdiscs. **d** The color and photoluminescence of solutions immediately after ultrasonication and after 12-h incubation under room light and UV light. **e** Bright-field and fluorescence images of dual-phase $Cs_4PbBr_6/CsPbBr_3$ microdiscs.

the solution onto a glass substrate and examined the sample using bright-field optical microscopy (Fig. 4c). Under the microscope, we observed that $Cs_4PbBr_6$ microdiscs appeared as $CsPbBr_3$ micro-cuboids were dissolving (Supplementary Movie 2). The formation of $Cs_4PbBr_6$ can be described as:

$$PbBr_4^{2-}(sol) + 2Br^-(sol) \rightarrow PbBr_6^{4-}(sol)$$

$$PbBr_6^{4-}(sol) + 2Cs^+(sol) \rightarrow Cs_4PbBr_6(s)$$

The fluorescence quantum yield of the microdiscs is nearly zero immediately after their formation, but gradually increases over time (Fig. 4d). This is due to the conversion of single-phase $Cs_4PbBr_6$ to dual-phase $Cs_4PbBr_6/CsPbBr_3$ (step vi). This conversion occurs spontaneously at room temperature, but at a much slower speed over 12 h. The final $Cs_4PbBr_6/CsPbBr_3$

microdiscs emit bright fluorescence (Fig. 4e). The conversion of single-phase $Cs_4PbBr_6$ microdiscs proceeds with slow self-formation of $CsPbBr_3$ NCs in the $Cs_4PbBr_6$ matrix, releasing CsBr to the solution, via:

$$Cs_4PbBr_6(s) \rightarrow (1-x)Cs_4PbBr_6/xCsPbBr_3(s) + 3xCsBr(sol)$$

where $x$ («1) denotes the amount of conversion. A similar CsBr extraction process has previously been observed during the evolution of single-phase $Cs_4PbBr_6$ to single-phase $CsPbBr_3$[7,25].

The sonochemical synthesis of dual-phase $CsPb_2Br_5/CsPbBr_3$ composite ($a = 1$, $b = 3$) appeared to be straightforward without producing any apparent intermediates. Time-lapse video (Supplementary Fig. 20, Supplementary Movie 3) shows that as soon as the precursor salts are placed in DMF, orange-color $CsPbBr_3$ layers are formed on the surface of the salts. After 2 min of ultrasonication,

the entire solution turns yellow as $CsPb_2Br_5/CsPbBr_3$ microparticles are produced. This fast and simple formation is in contrast to the slow formation of $Cs_4PbBr_6/CsPbBr_3$. Considering the tetragonal structure of $CsPb_2Br_5$ with alternating $Cs^+$ and $Pb_2Br_5^-$ layers, the formation mechanism may be described as:

$$2PbBr_2(s) + Br^-(sol) \rightarrow Pb_2Br_5^-(sol)$$

$$Pb_2Br_5^-(sol) + Cs^+(sol) \rightarrow CsPb_2Br_5(s)$$

During the growth of $CsPb_2Br_5$, the self-formation of $CsPbBr_3$ NCs in the $CsPb_2Br_5$ matrix simultaneously occurs by releasing $PbBr_2$ to the solution:

$$CsPb_2Br_5(s) \rightarrow (1-x)CsPb_2Br_5/xCsPbBr_3(s) + xPbBr_2(sol).$$

The release of $PbBr_2$ during the self-formation of NCs promotes the formation of the $CsPb_2Br_5$ matrix.

In summary, we have shown that the sonochemical processes led to rapid synthesis of dual-phase perovskites. Both $Cs_4PbBr_6/CsPbBr_3$ and $CsPb_2Br_5/CsPbBr_3$ composites have well-defined endotaxy structures with good lattice matching between embedded $CsPbBr_3$ NCs and the non-luminescent matrices[4,9,26]. The high solid-state PLQY of >40% in $Cs_4PbBr_6/CsPbBr_3$ and efficient lasing from single $CsPbBr_3$ microparticles as small as 2 μm attest the high quality of the microcrystals. Lastly, our real-time measurement data suggest that $CsPbBr_3$ NCs in the $Cs_4PbBr_6$ or $CsPb_2Br_5$ matrix is formed via a partial extraction of CsBr or $PbBr_2$. Single- and dual-phase $CsPbBr_3$-based microparticles may prove to be useful building blocks for optical devices.

## Methods

**Chemicals and reagents**. CsBr (99.99%), $PbBr_2$ (99.99%), N,N-dimethylforma-mide (anhydrous, 99.8%), acetone (99.9%), ethyl acetate (EtoAC) (anhydrous, 99.8%), γ-butyrolactone (GBL) (99.9%), and dimethyl sulfoxide (DMSO) (anhydrous, 99.9%) were purchased from Sigma-Aldrich. All reagents were used as received from Sigma-Aldrich without further purification.

**Sonochemical synthesis of single-phase perovskite microcrystals**. For producing inorganic perovskite $CsPbBr_3$, CsBr and $PbBr_2$ were dispersed at an equal concentration in 1 mL of N,N-dimethylformamide (DMF) in a vial. The typical concentration was 0.075 M (i.e., $a = 1$ and $b = 1$) or its multiples up to 0.3 M ($a = b = 2$, 3, or 4). The vial was placed into a bath-type ultrasonicator (Elmasonic P60H, Elma) or a single-step tip ultrasonicator (Fisherbrand Q125) in room temperature and irradiated with ultrasonic waves (frequency: 20 kHz~80 kHz). After 2–3 min of ultrasonication, single-phase $CsPbBr_3$ microcrystals were spontaneously crystallized and dispersed in the solution.

**Sonochemical synthesis of dual-phase $Cs_4PbBr_6/CsPbBr_3$ microcrystals**. Ultrasonication of 1 mL of DMF solution of CsBr (0.225 M or 0.3 M) and $PbBr_2$ (0.075 M) for 2 min yields single-phase $CsPbBr_3$ microcrystals. Continuing ultrasonication for additional several min makes the orange-colored solution to white, opaque dispersion of single-phase $Cs_4PbBr_6$ microparticles, and then to lemon-colored solution of dual-phase $Cs_4PbBr_6/CsPbBr_3$ microparticles. This process takes about 13–15 min. After the ultrasonication has been stopped, the color of the solution becomes gradually brighter at room temperature overnight. The morphologies of the final particles are a mixture of hexagon and rhombus. As an alternative way to synthesize dual-phase $Cs_4PbBr_6/CsPbBr_3$ microparticles, after the synthesis of $CsPbBr_3$ microcrystals by 2 min of ultrasonication, the solution is removed from the ultrasonicator, and then vigorous shaking is applied for 1 h until the color turns to light green. Likewise, the lemon color becomes gradually intense over time in room temperature.

**Sonochemical synthesis of single-phase $Cs_4PbBr_6$ microcrystals**. For single-phase $Cs_4PbBr_6$, CsBr (0.075 M), and $PbBr_2$ (0.0375 M) in 1 mL of DMF were used as a precursor solution. In the case of CsBr (0.15 M) and $PbBr_2$ (0.075 M), a mixture of single-phase $CsPbBr_3$ particles and $Cs_4PbBr_6$ particles were formed.

**Sonochemical synthesis of dual-phase $CsPb_2Br_5/CsPbBr_3$ microcrystals**. Dual-phase $CsPb_2Br_5/CsPbBr_3$ were obtained when the concentration of $PbBr_2$ was higher than the concentration of CsBr. Regular truncated octahedron morphology was obtained with CsBr (0.075 M) and $PbBr_2$ (0.15 M). Cuboctahedron particles were obtained with CsBr (0.075 M) and $PbBr_2$ (0.225 M). For wedding cake or fibrous structure with rough surface, CsBr (0.075 M) and $PbBr_2$ (0.3 M) were used.

**Structural characterization**. For SEM and EDX measurements, LHPs micro-crystals were transferred onto a chipped Si wafer by drop-casting and imaged using a Zeiss Merlin high-resolution SEM equipped with an EDX detector operated at 15 kV. For TEM measurements, samples were prepared by drop-casting LHP microparticles onto TEM grids (Ted Pella). TEM images were acquired using a FEI Tecnai Multipurpose TEM operated at 120 kV. The illumination beam was expanded to avoid sample damage. For PXRD measurements, PXRD patterns over $2\theta$ angles from 10° to 60° were collected using a PANalytical X'Pert PRO high-resolution X-ray diffraction system with a CuKα irradiation source. These measurements were performed at MIT Center for Material Science and Engineering (CMSE).

**Optical characterization**. See Supplementary Methods.

**Reporting summary**. Further information on research design is available in the Nature Research Reporting Summary linked to this article.

## Data availability

The main data supporting the finding of this study are available within the paper and its Supplementary Information file. Other relevant data are available from the corresponding author upon reasonable request.

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

## Acknowledgements

This research was supported in part by the National Institutes of Health (grant no. DP1EB024242). S.C. acknowledges the Samsung Scholarship. Part of this work used the facilities in the Center for Materials Science and Engineering at MIT.

## Author contributions

S.C. and S.H.Y. conceived the idea and designed the experiments. S.C. performed experiments and analyzed the data. S.H.Y. supervised the project. Both authors wrote the paper.

## Competing interests

The authors declare no competing interests.
