## [Peer Review File · Communications Chemistry]

Reviewers' comments:

Reviewer #1 (Remarks to the Author):

The authors have performed the synthesis of CsPbBr₃ perovskite nanoparticles embedded in 0D (Cs₄PbBr₆) and 2D (CsPbBr₅) cesium lead halide matrices with the aim to determine the structure of the composites in-depth as well as the origin of their green photoluminescence, what is currently considered a controversial issue.

The preparation of CsPbBr₃ perovskite, as well as the dual-phase CsPbBr₃/Cs₄PbBr₆ and CsPbBr₃/CsPbBr₅ microcrystals, have been prepared by sonochemistry. The formation kinetics of the microcrystals as well as the phase transition of the cesium lead halide microcrystals have been studied comprehensively.

They demonstrate that the CsPbBr₃ nanocrystals of the composites are the actual photoluminescent sites and suggest that the high photoluminescence quantum yield of CsPbBr₃/Cs₄PbBr₆ microcrystals comes from the low dielectric constant of the Cs₄PbBr₆ matrix. In addition, the failure to observe lasing in the CsPbBr₃/Cs₄PbBr₆ and CsPbBr₃/CsPbBr₅ microcrystals is associated to the low refractive index of the matrices in which the CsPbBr₃ nanocrystals are embedded.

In my opinion, this work is convincing and deserves publication in Communications Chemistry due to its importance to researchers in the perovskite field.

Minor revisions

- Some data for producing organic-inorganic hybrid perovskites have been included in the Methods section. This information is not related with the herein reported systems and should be removed.
- The Discussion section is in fact the Conclusions section.

Reviewer #2 (Remarks to the Author):

The manuscript "Structure and optical properties of perovskite-embedded dual-phase microcrystals synthesized by sonochemistry" by Sangyeon Cho et al., reports the structure and optical results of CsPbBr₃, Cs₄PbBr₆/CsPbBr₃, and CsPb₂Br₅/CsPbBr₃ microcrystals. Based on the comments below I find the manuscript lacking strong evidences for its conclusion.

1. I do not agree with this statement "While much progress in the dual-phase materials is expected, it has been difficult to visualize the relative lattice orientation between LHPs NCs and the host matrix clearly by using HRTEM, due to difficulties such as the low damage threshold of the materials by electron beam and the inadequate sample sizes being too small (nano-sizes) or too large (centimeter-sizes)". Cs₄PbBr₆ NCs are reported with sizes up to 120 nm (Yin et al. ACS Energy Lett. 2017, 2, 2805-2811). In fact, high quality HRTEM images requires nm-scale sizes/thickness.

2. Thick samples such as microcrystals reported in this manuscript are very difficult to analyze by HRTEM. The quality of the HRTEM images provided here is very low and it is difficult to validate the soundness of the results.

3. One of the main claims/findings of this manuscript is the precise relative lattice orientation of the endotaxial structures. Authors should provide FFT analysis to confirm their findings.

4. In Figure 2a, how can the authors distinguish between the two 3.7 Å lattice spacing assigning one to (212) Cs₄PbBr₆ and the other to (210) CsPbBr₃? In figure 2c, a d-spacing of 5.8 Å is in a very good match with the (110) of CsPbBr₃. Not clear why the authors assign it to the CsPb₂Br₅

phase.

5. The authors suggest that the CsPbBr₃ NCs inclusion in the Cs₄PbBr₆/CsPbBr₃ sample to be around 4.2 nm. However, they show that this sample is emitting at 512 nm. According to Almeida et al. (ACS Nano 2018, 12, 1704-1711), 4 nm CsPbBr₃ emits at 470-480 nm.

6. Based on the histogram, the size of the CsPbBr₃ inclusions is actually even less than 10 and larger than 20 nm. In addition, how do the authors determine the diameter of these very irregular shapes?

7. Mechanism of dual-phase formation is somehow confusing. Page 24 Line 188 authors report that as CsPbBr₃ microcuboids were slowly dissolving, Cs₄PbBr₆ microdiscs appeared near the sites of dissolution. This sounds like to individual processes which does not match the formation of Cs₄PbBr₆/CsPbBr₃ in the TEM images. Moreover, Line 199, the authors suggest self-formation of CsPbBr₃ NCs in the Cs₄PbBr₆ matrix. What triggers this self-formation?

8. I find parameters 'a' and 'b' used by the authors to denote the concentrations of the CsBr and PbBr₂ precursors very complicated.

9. Is the dual phase still of a potential interest for micro-lasers and light-emitting diodes as the author report in the conclusion despite dual-phase microparticles did not support laser oscillation even at higher pump energy levels as the authors reported in the results section. What about the insulating characteristic of the Cs₄PbBr₆ and CsPb₂Br₅?

Further comments:

1. Is CsPb₂Br₅ really considered as a Ruddlesden-Popper phase as the authors suggest in the introduction?

2. Page 2 Line 60: DMF provided best results in terms of what exactly. What were the other explored solvents?

3. Page 2 Line 62: how did the authors confirm it is a colloidal solution? Authors should perform DLS analysis.

4. XRD of the single phase Cs₄PbBr₆ is missing.

5. What is the PLQY of CsPb₂Br₅/CsPbBr₃?

6. In the experimental section the authors report the synthesis of CH₃NH₃PbBr₃, yet there is no any discussion related to this perovskite in the manuscript.

7. Can the authors provide similar dual-phase microcrystals for the other halides (Cl and/or I)?

Point by Point Response

We would like to thank the editor and reviewers for their comments and constructive suggestions. We carefully considered each and every point and made changes to address the concerns. Our point-by-point responses – Author Response (AR) – are as follows.

Reviewer #1 (Remarks to the Author):

The authors have performed the synthesis of CsPbBr₃ perovskite nanoparticles embedded in 0D (Cs₄PbBr₆) and 2D (CsPbBr₅) cesium lead halide matrices with the aim to determine the structure of the composites in-depth as well as the origin of their green photoluminescence, what is currently considered a controversial issue.

The preparation of CsPbBr₃ perovskite, as well as the dual-phase CsPbBr₃/Cs₄PbBr₆ and CsPbBr₃/CsPbBr₅ microcrystals, have been prepared by sonochemistry. The formation kinetics of the microcrystals as well as the phase transition of the cesium lead halide microcrystals have been studied comprehensively.

They demonstrate that the CsPbBr₃ nanocrystals of the composites are the actual photoluminescent sites and suggest that the high photoluminescence quantum yield of CsPbBr₃/Cs₄PbBr₆ microcrystals comes from the low dielectric constant of the Cs₄PbBr₆ matrix. In addition, the failure to observe lasing in the CsPbBr₃/Cs₄PbBr₆ and CsPbBr₃/CsPbBr₅ microcrystals is associated to the low refractive index of the matrices in which the CsPbBr₃ nanocrystals are embedded.

In my opinion, this work is convincing and deserves publication in Communications Chemistry due to its importance to researchers in the perovskite field.

AR: We thank the reviewer for seeing the potential of our work to be an important resource for further studies in the perovskite field.

Minor revisions

- Some data for producing organic-inorganic hybrid perovskites have been included In the Methods section. This information is not related with the herein reported systems and should be removed.

AR: We agree with the reviewer's suggestion.

Changes made: We removed the contents about organic-inorganic hybrid perovskites in this manuscript.

- The Discussion section is in fact the Conclusions section.

AR: Thank you for this comment.

Changes made: We changed the name of the last section from 'Discussion' to 'Conclusion'.

Reviewer #2 (Remarks to the Author):

The manuscript " Structure and optical properties of perovskite-embedded dual-phase microcrystals synthesized by sonochemistry" by Sangyeon Cho et al., reports the structure and

optical results of CsPbBr_3 , $\text{Cs}_4\text{PbBr}_6/\text{CsPbBr}_3$, and $\text{CsPb}_2\text{Br}_5/\text{CsPbBr}_3$ microcrystals. Based on the comments below I find the manuscript lacking strong evidences for its conclusion.

1. I do not agree with this statement “While much progress in the dual-phase materials is expected, it has been difficult to visualize the relative lattice orientation between LHPs NCs and the host matrix clearly by using HRTEM, due to difficulties such as the low damage threshold of the materials by electron beam and the inadequate sample sizes being too small (nano-sizes) or too large (centimeter-sizes)”. Cs_4PbBr_6 NCs are reported with sizes up to 120 nm (Yin et al. ACS Energy Lett. 2017, 2, 2805-2811). In fact, high quality HRTEM images requires nm-scale sizes/thickness.

AR1: We apologize for not making it clear that the samples we used for HRTEM were indeed thin with < 100 nm of thickness. Representative SEM images of these samples are shown below. We used sheet-like $\text{Cs}_4\text{PbBr}_6/\text{CsPbBr}_3$ with an average thickness of 80 nm (15 samples) and wedding-cake-like $\text{CsPb}_2\text{Br}_5/\text{CsPbBr}_3$ consisting of multiple thin layers each with an average thickness of 50 nm (15 samples). Each sample was a few μm in lateral dimension, so we were able to image both the matrix and nanocrystals, acquiring total four different lattice fringes, from which the relative lattice orientations were determined.

Changes made: We included this figure in **Supplementary Fig. S13**. On page 2, we revised that “due to difficulties such as the low damage threshold of the materials by electron beam and the inadequate sample sizes being too small lateral size (less than 10 nm) to observe more than four different lattice planes at given electron beam or too large in thickness (larger than few hundreds of nm) in thickness to get clear images^{7,12-14}”.

2. *Thick samples such as microcrystals reported in this manuscript are very difficult to analyze by HRTEM. The quality of the HRTEM images provided here is very low and it is difficult to validate the soundness of the results.*

AR2: Please see our response AR1. We indeed used nanometer-scale laminar structures that are appropriate for HRTEM (Fig. S13). The lattice structures and diffraction patterns we obtained from the HRTEM images were sufficient to allow us to determine relative lattice orientations between the endotaxy matrix and the CsPbBr₃ nanocrystals. Please also see figure below.

Changes made: We added sentences to clarify the sample conditions for HRTEM imaging in page 3. We wrote: “Unlike previous dual-phase materials^{7,12–14}, structurally anisotropic microparticles produced by sonochemistry were well suited to obtain high-quality HRTEM having more than four different lattice planes at given the direction of an electron beam. For HRTEM imaging, we used micro-discoidal Cs₄PbBr₆/CsPbBr₃ (a=3, b=1) having average thickness of 80 nm (N=15) and wedding-cake CsPb₂Br₅/CsPbBr₃ (a=1, b=4) consisting of multiple thin layers having average thickness of 50 nm (N=15) (Supplementary Fig. 13).”

3. *One of the main claims/findings of this manuscript is the precise relative lattice orientation of the endotaxial structures. Authors should provide FFT analysis to confirm their findings.*

AR3: As requested, we provide our FFT analysis data below. In the case of Cs₄PbBr₆/CsPbBr₃, we assigned the well-defined lattice fringes with 3.7 Å and 4.2 Å at an intersection angle of 42° to the (212) and (210) axes of the Cs₄PbBr₆ matrix, respectively, and assigned the high-contrast lattice fringes with 3.7 Å and 3.5 Å at an intersection angle of 67 ° to the (210) and (021) axes of the CsPbBr₃ NCs, respectively. The [210] axis of Cs₄PbBr₆ is tilted by 6° with respect to the [021] axis of CsPbBr₃. For CsPb₂Br₅/CsPbBr₃, the low-contrast lattice fringes with 2.6 Å and 5.8 Å having an intersection angle of 35° were assigned to the (222) and (110) axes of the CsPb₂Br₅ matrix, respectively, and the lattice fringes in darker sub-regions with 6.8 Å and 11.6 Å having an intersection angle of 35° were assigned to the (210) and (021) axes of the CsPbBr₃ NCs. Hence, the [110] axis of CsPb₂Br₅ is aligned to the [121] axis of CsPbBr₃.

Changes made: We included this additional data in Supplementary Fig. S14. To explain the FFT analysis, we added in page 3: “The real-space images and the corresponding fast Fourier transform (FFT) analysis revealed the relative lattice orientation of the endotaxial structures (Figs. 2a & 2c and Supplementary Fig. 14). In the case of Cs₄PbBr₆/CsPbBr₃, well-defined lattice fringes with 3.7 Å and 4.2 Å having intersection angle of 42° are indexed to (212) and (210) of the Cs₄PbBr₆ matrix and higher-contrast lattice fringes with 3.7 Å and 3.5 Å having intersection angle of 67 ° are indexed to (210) and (021) of the CsPbBr₃ NCs. This suggests that the [210] axis of Cs₄PbBr₆ is tilted by 6° with respect to the [021] axis of CsPbBr₃. For CsPb₂Br₅/CsPbBr₃, low-contrast lattice fringes with 2.6 Å and 5.8 Å having intersection angle of 35° are indexed to (222) and (110) of the CsPb₂Br₅ matrix and lattice fringes in darker sub-regions with 6.8 Å and 11.6 Å having intersection angle of 35° are indexed to (210) and (021) of the CsPbBr₃ NCs. Hence, the [110] axis of CsPb₂Br₅ is aligned to the [121] axis of CsPbBr₃”.

4. In Figure 2a, how can the authors distinguish between the two 3.7Å lattice spacing assigning one to (212) Cs₄PbBr₆ and the other to (210) CsPbBr₃? In figure 2c, a d-spacing of 5.8Å is in a very good match with the (110) of CsPbBr₃. Not clear why the authors assign it to the CsPb₂Br₅ phase.

AR4: Please refer to our response AR3 above and Fig. S14. If we assign a lattice fringe having d-spacing of 5.8 Å to (110) of CsPbBr₃, the intersection angle between the corresponded fringe and other fringes in CsPbBr₃ NCs is 76° with (111) plane and 36° with (101) plane, which is not consistent with theoretical estimations (35° with (111) and 60° with (101) plane). On the contrary, the current assignment of the fringe to (110) of CsPb₂Br₅ shows good agreement in both d-spacing and the interaction angle with the other lattice fringes as shown in Fig. S14.

5. The authors suggest that the CsPbBr₃ NCs inclusion in the Cs₄PbBr₆/CsPbBr₃ sample to be around 4.2 nm. However, they show that this sample is emitting at 512 nm. According to Almeida et al. (ACS Nano 2018, 12, 1704-1711), 4 nm CsPbBr₃ emits at 470-480 nm.

AR5: Please see the section ‘Optical properties of dual-phase LHPs’ on Page 4 in the main text. The 4-nm CsPbBr₃ NC_s in Almeida et al. is *not* endotaxially-embedded NCs but conventional NCs passivated by lipid molecules such as oleyl acids and oleyl amines. The emission of NCs is influenced by their surrounding dielectric media. We explained this discrepancy by using quantum confinement (See Eq. (1)). Note that our measurement presents a good agreement with former reports. Our 4.2-nm ($\sigma=0.7$ nm) CsPbBr₃ NCs in Cs₄PbBr₆ have emission at 510~521 nm (See Supplementary Fig. 16). According to Xu et al. [Adv. Mat. 2017, 29, 1703703], 4~8 nm sized CsPbBr₃ embedded in a thin film of Cs₄PbBr₆ emits fluorescence ranging from 510 to 520 nm. According to Chen et al. [Adv. Func. Mat. 2018, 28, 1706567], 2.85-nm-sized ($\sigma=0.75$ nm) CsPbBr₃ NCs embedded in Cs₄PbBr₆ emits at 520 nm.

6. Based on the histogram, the size of the CsPbBr₃ inclusions is actually even less than 10 and larger than 20 nm. In addition, how do the authors determine the diameter of these very irregular shapes?

AR6: Thanks for this question. In fact, we defined the effective size of a NC to be the diameter of the largest circle that circumscribes the NC. The size of CsPbBr₃ NCs embedded in CsPb₂Br₅ was $D=14.6$ nm with $\sigma=4.6$ nm. This relatively large variation is thought to originate from the rapid formation kinetics of dual-phase CsPb₂Br₅.

Changes made: To make this point clear, we added sentences to explain the measurement of effective size of embedded CsPbBr₃ NCs in page 4. We wrote: “From the TEM images, we determined the effective size of CsPbBr₃ NCs embedded in the matrices by measuring the diameter of the largest circle circumscribing the NCs (Supplementary Fig. 15).”

7. Mechanism of dual-phase formation is somehow confusing. Page 24 Line 188 authors report that that as CsPbBr₃ microcuboids were slowly dissolving, Cs₄PbBr₆ microdiscs appeared near the sites of dissolution. This sounds like to individual processes which does not match the formation of Cs₄PbBr₆/CsPbBr₃ in the TEM images. Moreover, Line 199, the authors suggest self-formation of CsPbBr₃ NCs in the Cs₄PbBr₆ matrix. What triggers this self-formation?

AR7: Please see the section ‘Mechanism of dual-phase formation’ on page 4 of the main text. The formation mechanism of Cs₄PbBr₆/CsPbBr₃ can be summarized as follows: First, Cs₄PbBr₆ microdiscs is formed as CsPbBr₃ microcuboids are dissolved; the generated Cs₄PbBr₆ microdiscs are non-luminescent, but over time CsPbBr₃ inclusions are formed, which makes the Cs₄PbBr₆/CsPbBr₃ able to generate green luminescence and the luminescent intensity increases as more CsPbBr₃ NCs are formed. The precise mechanisms for the formation and growth of CsPbBr₃ NCs in the endotaxy matrix are not fully understood [See Akkerman et al. [J. Phys. Chem. Lett., 2018, 9 2326-2337]. Based on this and previous observations [Palazon et al. ACS. Energy. Lett. 2017, 2, 2445-2448], we have postulated that the intrinsic thermodynamic instability of Cs₄PbBr₆ in the saturated precursor solution triggers post-extraction of CsBr, which then leads to the formation of endotaxial CsPbBr₃ NCs. However, verification of this hypothesis is beyond the scope of this manuscript.

Changes made: To explain the formation mechanism, we added a sentence in page 6: “Under the microscope, we observed that Cs₄PbBr₆ microdiscs appeared as CsPbBr₃ microcuboids were dissolving (Video S2).”

8. I find parameters ‘a’ and ‘b’ used by the authors to denote the concentrations of the CsBr and PbBr₂ precursors very complicated.

AR8: Thanks for your comment. We made efforts to address this concern.

Changes made: We went through the manuscript to cite the actual concentrations instead of the parameters 'a' and 'b'. However, we used these parameters to denote the concentration ratios.

9. Is the dual phase still of a potential interest for micro-lasers and light-emitting diodes as the author report in the conclusion despite dual-phase microparticles did not support laser oscillation even at higher pump energy levels as the authors reported in the results section. What about the insulating characteristic of the Cs₄PbBr₆ and CsPb₂Br₅?

AR9: Thanks for this comment. Cs₄PbBr₆ and CsPb₂Br₅ have wider bandgap 3.1 eV and 3.7 eV, both larger than the 2.31 eV band gap energy of CsPbBr₃. So, the matrix could serve as an insulator. As we described on Page 5, the failure to reach lasing threshold is attributed to the weak confinement of the optical mode within the microparticles because of the lower refractive index (~1.8) of the Cs₄PbBr₆ and CsPb₂Br₅ matrices. The low index serves as a disadvantage for laser generation but can be an advantage for luminescent devices for light extraction. We made changes to clarify this point.

Changes made: We made the following revision on page 5: "The smallest size of lasing CsPbBr₃ microcrystals was 2 μm (Fig. 3b and Supplementary Fig. 18). The laser emission linewidth was approximately 0.2 nm above a threshold pump energy of 1.7 mJ/cm², and the spontaneous emission factor was 0.05 (Fig. 3c). The other device presents the laser emission linewidth of 0.3 nm above a threshold pump energy of 2.2 mJ/cm², and the spontaneous emission factor was 10⁻³ (Supplementary Fig. 18). The converted threshold pump energy to threshold carrier density is about ~3 x 10¹⁹ cm⁻³, which is higher than theoretical estimation of Mott density (10¹⁸ cm⁻³)¹⁹. This infers lasing in electron hole plasma state (EHP) rather than excitonic state^{5,19}, which is beneficial to build up large population inversion via band-gap renormalization (BGR)²⁰. The laser emission of single-phase CsPbBr₃ microparticles in air (15 samples) at a pump fluence twice the lasing threshold was prolonged for 10⁵ pump pulses (5,000 s at 20 Hz) with a pulse-to-pulse wavelength fluctuation of 0.47 nm (Supplementary Fig. 18)..... Considering the high PLQY of CsPbBr₃ NCs in the dual-phase composites, we attribute the failure to reach lasing threshold to weaker cavity resonance due to the lower refractive index (~1.8) of the Cs₄PbBr₆²¹ and CsPb₂Br₅ matrices compared to the index (~2.6) of CsPbBr₃⁵ and relatively small amount of the optical gain in the excitonic state due to un-normalized band gap."

Further comments:

1. Is CsPb₂Br₅ really considered as a Ruddlesden-Popper phase as the authors suggest in the introduction?

AR1: Thanks for your comment. We realized that it was misnomer. CsPb₂Br₅ has a two-dimensional structure with Pb₂Br₅⁻ layers alternating with Cs⁺ cations. Because the typical chemical stoichiometry of Ruddlesden-Popper (RP) phase is A₂PbX₄ and CsPb₂Br₅ structure made of layers of corner-sharing [PbX₆]⁴⁻ octahedra spaced by cations [Akkerman et. al., Nano Lett., 2017, 17, 1924-1930], CsPb₂Br₅ may not be regarded as the RP phase, but one of general quasi-two-dimensional (2D) phases [C. Qin et. al., J. Phys. Chem. Lett. 2017, 8, 21, 5415-5421 & X. Zhang et. al., ACS Appl. Mater. Interfaces, 2018, 10, 8, 7145-7154].

Changes made: We replaced the term of 'Ruddlesden-Popper' by 'quasi-two-dimensional (2D)'.

2. Page 2 Line 60: DMF provided best results in terms of what exactly. What were the other explored solvents?

AR2: We have tested several other polar aprotic solvents with different dipole moments (D), such as Acetone (2.91D), ethyl acetate (EtoAC; 1.78D), γ -butyrolactone (GBL; 4.244D), and dimethyl sulfoxide (DMSO; 3.96D). The photos of the products synthesized with concentrations of $a = 75$ mM and $b = 75$ mM are shown in Supplementary Fig. S2 below. Among the various solvents we have tested, DMF was the only one that yielded orthorhombic CsPbBr_3 microcrystals with green luminescence.

Changes made: We included the above information and added the figure as Supplementary Fig. S2. On Page 2, we wrote: "For CsBr and PbBr_2 salts, N,N-dimethylformamide (DMF) produced high-quality CsPbBr_3 microparticles among different polar aprotic solvents having similar dipole moments (acetone, ethyl acetate (EtoAC), γ -butyrolactone (GBL) and dimethyl sulfoxide (DMSO)) (Supplementary Fig. 2)."

3. Page 2 Line 62: how did the authors confirm it is a colloidal solution? Authors should perform DLS analysis.

AR: We agree with your statement, and thanks for pointing this out. The microcrystals sediment after the ultrasonication. For this reason, DLS analysis could not be performed. We corrected this error.

Changes made: We have removed the work 'colloidal' in the main text.

4. XRD of the single phase Cs_4PbBr_6 is missing.

AR: Thanks for this comment. The XRD data of single phase Cs_4PbBr_6 are shown in Supplementary Fig. S9 below.

Changes made: We included the XRD data as Supplementary Fig. S9.

5. What is the PLQY of $CsPb_2Br_5/CsPbBr_3$?

AR5: The measured PLQY of $CsPb_2Br_5/CsPbBr_3$ is 0.16%, which is presented in Table S1.

6. In the experimental section the authors report the synthesis of $CH_3NH_3PbBr_3$, yet there is no any discussion related to this perovskite in the manuscript.

AR1: Thanks for your comment. To focus on CsPbBr₃, we removed the description about CH₃NH₃PbBr₃ from the experimental section.

Changes made: We have removed the contents about CH₃NH₃PbBr₃ perovskites.

7. Can the authors provide similar dual-phase microcrystals for the other halides (Cl and/or I)?

AR7: Thanks for this suggestion. It is technically possible to form dual-phase microcrystals for other halides, such as Cl and I, by post-synthetic anion exchange of Br. However, the perovskite α -phase of CsPbI₃ is thermodynamically unstable due to the large diameter of iodine ions. Therefore, this material is going to be readily transformed into non-perovskite δ -phase [Z. Li *et al.*, Chem. Mater., 2016, 28, 1, 284-292]. This is potential topic for future research.

REVIEWERS' COMMENTS:

Reviewer #2 (Remarks to the Author):

The authors have responded to all comments/questions. The manuscript is recommended for publication.